# Excess Mortality of Males Due to Malignant Lung Cancer in OECD Countries

**DOI:** 10.3390/ijerph18020447

**Published:** 2021-01-08

**Authors:** Waclaw Moryson, Barbara Stawinska-Witoszynska

**Affiliations:** Department of Epidemiology and Hygiene, Social Medicine, Poznan University of Medical Sciences, Rokietnicka 4, 60-806 Poznan, Poland; bwitoszynska@ump.edu.pl

**Keywords:** excess mortality of males, lung cancer, OECD, smoking

## Abstract

Excess mortality of men has been observed since the beginning of the 20th century. One of the main causes of this phenomenon is malignant cancers, with lung cancer as the main reason. At the turn of the 20th and 21st centuries, a decline in male excess mortality was observed in most developed countries. This study aimed to analyze the changes in the level of excess mortality of men caused by lung cancer between 2002 and 2017 in the countries associated with the Organization for Economic Cooperation and Development (OECD). In order to compare changes in male mortality rates across countries, the annual average percent change (AAPC) in male excess mortality rate for a given country was calculated. A decrease in excess male mortality due to lung cancer between 2002 and 2017 was recorded in 33 of the 35 countries analyzed. The highest rate of decline was observed in Spain (4.9% per year), Belgium (4.7% per year), Slovakia (4.4% per year) and other European OECD member countries. In most OECD countries, the decrease in excess male mortality was the result of a decrease in mortality in the male population as well as the increase in female mortality. In 10 member countries of the OECD, there was a simultaneous increase in the mortality of men and women, though the increase was higher in the female population. Only in two countries was the decrease in excess mortality of males due to a greater decline in the mortality of men. The decline in male mortality from lung cancer primarily associated with a reduction in tobacco consumption in this population initiated in the 1980s or 1990s (depending on the country) was not observed in the female population.

## 1. Introduction

Presently, the average male life expectancy at birth is shorter than that of females [1]. This phenomenon is caused by the excess of male deaths over female deaths. Life expectancy of females reached disproportionate levels with respect to that of males at the beginning of the 20th century; however, at that time, in most countries, women were expected to live on average for only about two years longer than men [2]. Over the following decades, the disparities in life expectancy between men and women increased, especially after the Second World War [3]. According to global data available for the years 1950–2017, life expectancy for men aged 0 increased from 48.1 to 70.5 years, whereas life expectancy for women increased from 52.9 to 75.6 years [4]. In 2017, the difference in life expectancy between men and women worldwide was 5.1 years due to the disadvantage of men. The growth in male excess mortality was largely due to chronic diseases associated with increased male exposure to risk factors such as nicotinism, alcohol consumption and unhealthy eating habits [5]. One of the main causes of the excess mortality of males was malignant types of cancer, primarily lung cancer, which remains the main cause of male cancer deaths in 93 countries [6]. Epidemiological analyses published in recent years show that at the turn of the 20th and 21st centuries, a reduction in the gender gap in life expectancy was observed [4]. In 2017, the world’s most comprehensive observational epidemiological study to date, “Global Burden of Disease”, published an analysis of the mortality of men and women worldwide over a period of 67 years between 1950 and 2017. The results of this study indicate that the decline in male excess mortality in the best developed countries has been taking place since the 1990s [4]. Moreover, international health care institutions report that there has been a decrease in the exposure of men and women to risk factors associated with the development of the gender gap in mortality rates over recent decades [7,8]. There has also been a decline in mortality caused by cancer [9] (which has been the reason for higher mortality rate in the male population for many years).

The data gathered in the countries belonging to the Organization for Economic Cooperation and Development (OECD) show that the habit of smoking in both sexes has been decreasing, but this reduction occurred earlier in the male population and has thus reached a greater extent [8,9]. There has been a reduction in the exposure of men to harmful substances in the workplace, causing their mortality to be ten times higher than that of women. There has also been a marked reduction in the mortality of men due to cardiovascular diseases and cancer [9,10,11,12,13]. The aim of the study was to analyze changes in the level of excess mortality of men due to lung cancer between 2002 and 2017 in the countries of the OECD. We considered this topic to be important because the excess mortality of men concerning most diseases and deaths from external causes. However, despite the importance of the problem, the number of papers dealing with the subject seems to be inadequate. In view of increasing significance of cancers related deaths, the topic of excessive mortality of males due to lung cancer, the most common cause of death among cancer patients, is of great importance. This is especially so since previous reports on changes in lung cancer mortality in developed countries suggest changes in the socially and economically unfavorable phenomenon of excess mortality of males [14,15].

## 2. Materials and Methods

The data on male and female mortality caused by lung cancer were obtained from the International Agency for Research on Cancer (IARC) in 35 out of 36 OECD countries [16]. Due to the absence of information about Turkey, it was not included in the analysis. The data on male and female mortality included all deaths resulting from malignant cancer of the lung, bronchus and trachea, classified as C33-C34 according to the International Statistical Classification of Diseases and Related Health Problems (ICD-10), which were identified as the original cause of death [17].

In order to maintain the real dimension of the described phenomenon of excess mortality of men, in this publication the crude coefficients were used. However, to compare male and female mortality in the OECD countries, the mortality rates standardized based on World Standard Population were used [18].

In terms of available data for the period 2002–2017, using a linear regression model, the time trend of change in the crude mortality rates for women and men was determined for each of the analyzed countries. Based on the trends of changes in the abovementioned coefficients, the average annual percentage change (AAPC) of the coefficients was calculated together with the corresponding 95% confidence intervals (CIs).

Excess mortality of males was determined as the ratio of crude mortality rates in the male population to crude female mortality rates. It should be noted that the phrase “male excess mortality” may also have a different meaning. While in this work it means the excess deaths of men over deaths of women due to the same cause, the phrase may also mean the number of male deaths recorded in excess of the number of male deaths expected on the basis of past seasonal experience.

The overall percentage change in male excess mortality rates was determined for each of the analyzed countries as the percentage difference between the first and last available measures of the extent of the phenomenon between 2002 and 2017. As the OECD countries make the information on mortality available, covering a different time range between 2002 and 2017, the annual average percent change in male excess mortality rate (AAPC) for a given country was calculated as the next measure of change in male excess mortality caused by malignant cancer. A linear regression model was used to determine the AAPC of male excess mortality rates, as in the case of mortality trends.

In order to determine regression models, Joinpoint Regression statistical software version 4.7.0.0 (US National Cancer Institute) was used. The statistical significance level was assumed to be *p* < 0.05.

This work is designed as an epidemiological descriptive study. Its aim is to describe the trends in the excess mortality of men in OECD countries and to compare them with the trends in the prevalence of smoking in individual countries. The methodology used did not allow us to show a cause-and-effect correlation between these factors. For this reason, the juxtaposition of lung cancer mortality with the prevalence of smoking is based on the results of previous analytical studies, which are cited in the subsequent section of this paper.

## 3. Results

Different levels of lung cancer standardized mortality among men and women in different OECD countries are presented in Figure 1. The average standardized mortality rate for men in 2017 in OECD countries was 429.4/100,000, whereas for women it amounted to 14.7/100,000. The highest mortality rates were recorded in Hungary with 65.5/100,000 for men and 28.5/100,000 for women. Compared to the corresponding figures in Mexico (7.3/100,000 men, 3.8/100,000 women), they were approximately four times higher.

The lowest mortality rates as well as the lowest gender related differences were observed in the Scandinavian and South American countries. Lung cancer mortality rates were higher among men in all the countries analyzed except Iceland. The highest levels of male mortality were found in Eastern and Southern European countries, such as Hungary, Greece, Poland, Slovakia or Slovenia followed by the Asian OECD member states.

A decrease in male excess mortality caused by lung cancer between 2002 and 2017 was reported in 33 of the 35 countries analyzed, except for South Korea and Iceland (Figure 2 and Figure 3). In South Korea, the phenomenon of excess mortality in men increased by 4.1% between 2002 and 2016, resulting in a 2.9 times higher mortality rate in men than in women. Therefore, in the last year of analysis, the mortality rate in men due to lung cancer was 2.9 times higher in men than in women. In Iceland, on the other hand, throughout the entire analyzed period, the male and female mortality figures due to malignant pulmonary cancer remained at similar levels.

The order of OECD member countries according to the highest average annual rate of reduction in male excess mortality from lung cancer and the changes in both sexes’ mortality rates due to lung cancer is presented in Table 1.

The data contained in Table 1 indicate that in most countries the decrease in excess mortality of males due to lung cancer was the resultant of a decrease in mortality in the male population as well as the increase in female mortality. This pattern was observed in 23 of the OECD countries analyzed. In the remaining 10 countries, there was a simultaneous increase in the mortality of men and women, though it was higher in the female population. On the other hand, the only countries in which the decrease in excess mortality of males was due to a greater decline in the mortality of men than women were the United States and Mexico.

The average annual decline in male excess mortality rates (AAPC) was highest in Spain, where it amounted to 4.9%, Belgium (4.7%) and Slovakia (4.4%). In contrast, the overall percentage reduction in male excess mortality over the analyzed period proved to be the highest in Belgium, where it was 49%, in Spain 48.8% and in Luxembourg 47.8%.

Of the 17 OECD countries where the decrease in male excess mortality due to lung cancer proved to be the highest, the overall percentage reduction in male excess mortality between 2002 and 2017 exceeded one-third. All these countries are located in Europe, with seven of them in Central and Eastern Europe: Poland, Slovakia, the Czech Republic, Estonia, Lithuania and Hungary. Slovenia, the eighth Central European OECD country, also saw a dynamic decline in male excess mortality rate of 3.6% per year. However, due to an exceptionally low—contrary to the observed tendency—female mortality rate in the last year of analysis and at the same time an exceptionally high male mortality rate in that year, the overall reduction in excess mortality in this country was only −26.5%.

A similar trend was observed with regard to the average annual percentage reduction in male excess mortality rates. The nineteen countries with the fastest annual reductions in male excess mortality are those located in Europe. The average annual reduction in male excess mortality due to lung cancer ranges from 4.9% per year (Spain) to 2.4% per year (Greece) in these nineteen countries. The other European OECD countries (Sweden, Austria, Great Britain, Ireland, Denmark and Iceland) had low average annual reductions in male excess mortality, but these were the countries where this problem was on a relatively small scale—i.e., not exceeding 1.7. Portugal proved to be the only European OECD member state with significant lung cancer excess mortality of males (3.7), total percentage reduction in male excess mortality due to lung cancer not exceeding 20% and an annual average percentage reduction in male excess mortality not exceeding 2%.

It is noteworthy that all eight European countries of the former Eastern Bloc—among which at the turn of the 20th and 21st century the problem of excess mortality of males was particularly significant—were characterized by one of the fastest rates of decline of this phenomenon in the entire OECD.

A non-European country with the fastest rate of decline in excess mortality of males was New Zealand. Despite the small scale of male excess mortality due to lung cancer (in 2014 it amounted to 1.2), the scale of this phenomenon decreased by 22% between 2002 and 2014.

Korea, Japan and Israel were characterized by both a high excess male mortality rate and a slight improvement of this problem over the analyzed period. The male mortality rate due to malignant lung cancer in Korea was three times higher than among females. No significant change in this phenomenon was observed over the analyzed period. In Japan, the mortality rate among men was 2.6 times higher than that among women, and the total percentage reduction in male excess mortality caused by lung cancer was merely 8.6%.

In Israel, the mortality rate of men in the analyzed period was twice as high as in the female population, and the male excess mortality rate decreased by 14.2%.

In the nineteen countries with the highest reductions in excess mortality of males (more than 2.5% per year), there was an increase in female mortality and a decrease or small increase in male mortality of no more than 0.2%. In the countries with both a decrease in female and male mortality, the reduction in mortality was 2% per year in Mexico and 0.9% in the United States.

## 4. Discussion

In this paper, the prevalence of male excess mortality due to malignant lung cancer was analyzed. Based on the data from 2016, these cancers accounted for 25% of all male and 17% of female cancer deaths and were the main cause of death in OECD countries [9]. The mortality from these types of cancer is significantly higher than the global average in OECD countries. The average standardized lung cancer mortality rate for OECD countries in 2017 was 29.4/100,000 for men and 14.7/100,000 for women, while the global average mortality rates for the same year amounted to 27.1/100,000 for men and 11.2/100,000 for women [6]. These results are consistent with earlier observations of higher mortality from lung cancer in highly developed countries, which include all OECD countries studied except Mexico [19].

Based on lung cancer mortality rates which, in OECD countries as well as elsewhere in the world, are generally higher among men than women [6], the male excess mortality rate was calculated. The analysis of mortality trends contained in this publication demonstrated that the decrease in this phenomenon in most OECD countries results from a simultaneous increase in female mortality (33 out of 35 OECD countries) and a decrease in male mortality (25 out of 35 OECD countries). The results for the OECD countries are consistent with earlier reports of changes in mortality in several Western countries such as the United States, the United Kingdom, Canada and Australia. As early as 2010, Jemal A. et al. described the ongoing decline in male mortality and a plateau in female mortality since the end of the 20th century [14]. In addition, in their analysis of 2013, Lortet-Tieulent J. et al. described a decrease in the incidence of lung cancer among men in 14 out of 26 European countries in the age group of 35–64 years and in 15 countries in the age group of 65–74 years between 1998 and 2007. In the female population, on the other hand, they observed the stabilization of incidence of lung cancer among women in Eastern and Northern Europe, as well as an increase in lung cancer incidence in Western and Southern European countries [15]. This is partly due to lifestyle changes, including improved diet, reduced alcohol consumption as well as the reduced occupational exposure to carcinogens [9,10,11,12]. Above all, however, this is the result of changes in the consumption of tobacco by men and women [20,21]. Approximately 80% of deaths caused by malignant lung cancer are a consequence of smoking [22,23].

A model of the epidemic of nicotinism portraying the link between tobacco consumption and mortality from lung cancer described by A. D. Lopez is presented in Figure 4. The effect of a change in tobacco consumption in a given population, reflected in a change in mortality from malignant lung cancer, is visible with a delay of several decades [24].

Although tobacco consumption has been decreasing in all the OECD member states for many years now [7,25], the differences in the tendencies of excess mortality in these states are still noticeable and they mostly result from the varying course of the epidemic of nicotinism.

In North America, the United States and Canada, in Australia as well as in some of the countries of Western Europe, i.e., The United Kingdom, the Netherlands and Denmark, the prevalence of smoking among men peaked in the mid-20th century and since that time it has been steadily decreasing [7,26,27,28,29,30,31]. Consequently, the mortality rate of men has been decreasing in these countries since the 1990s, as it was the case between 2002 and 2017. In Central European countries, the reduction in tobacco consumption only began in the 1980s [7,32], followed by a period when the rate of smoking reduction was at a similar level as in the countries where tobacco reduction had started earlier [25,33]. Therefore, over the period from 2002 to 2017, roughly 20 years since smoking began declining in Central European countries and 50 years since the decline in smoking has been observed in North America, Australia and some of the countries of Western Europe, the reduction rates in male mortality in both groups remained at similar levels.

In the female population, the peak in the popularity of smoking took place later. In some countries, i.e., the United States, Canada, Australia, the United Kingdom, Denmark and Sweden, the highest prevalence of smoking took place in the 1980s and then decreased [7,25]. Nevertheless, a reduction in male mortality rates was observed in these countries. Despite a decrease in women’s tobacco consumption, the mortality caused by malignant pulmonary cancers dropped more slowly than in men or even increased.

In Italy, Germany, Spain and Switzerland, the course of the nicotinism epidemic among women was different. While tobacco consumption among men was decreasing, in the population of women it was stable, although lower than that of men [7]. This was reflected in a reduction in male excess mortality as a result of the simultaneous decline in mortality in men and its increase in women. In contrast, in countries such as Austria, France, Greece, Portugal and Lithuania, the incidence of nicotinism among women was increasing [7], reaching a higher level in the population of young women than in the population of men [25]. In the above countries, the decline in excess male mortality rate is mainly due to an increase in female mortality.

The changes in mortality observed in highly developed Asian countries are of particular interest. In Korea and Japan in the years 2002–2017, a stable level of male excess mortality was observed, with a simultaneous increase in female and male mortality. These changes were accompanied by a decrease in the popularity of smoking among men and a consistently low prevalence of smoking amongst women [7,34,35]. The analyses of mortality from lung cancer in these countries point out that the increase may result from starting to smoke at an ever younger age [34,36] which, historically speaking, had not been typical to this society [24]. Furthermore, the abovementioned countries saw an increasing incidence and mortality of women from nonsmoking related malignant pulmonary cancers. It was estimated that the share of these cancers among all diagnosed lung cancers in women in these countries nearly reached 79% [37].

It should be noted that in order to maintain the real dimension of the described phenomenon of excessive mortality of men, in this publication the crude coefficients were used, while in many quoted studies on lung cancer mortality, standardized coefficients were used.

The different systems of reporting cancer deaths across the countries under discussion constitute a limitation of this study. It should be noted, however, that almost all population cancer registries in the analyzed countries (34 out of 35, with the exception of Mexico) have been assessed by the WHO as high quality records [38].

## 5. Conclusions

Excess mortality of males due to lung cancer decreased between 2002 and 2017 in most OECD countries with the highest rate of decline in Spain, Belgium, Slovakia and other European OECD member states.

The reduction in the male population excess mortality rate resulted more often from a decrease in male mortality and an increase in female mortality, and less frequently from a greater decrease in male mortality than female mortality due to pulmonary cancer.

The decrease in the level of male mortality caused by malignant lung cancer is primarily related to the reduction in tobacco consumption in this population initiated in the 1980s or 1990s (depending on the country), which, according to many studies, was not observed in the female population.

## Figures and Tables

**Figure 1 ijerph-18-00447-f001:**
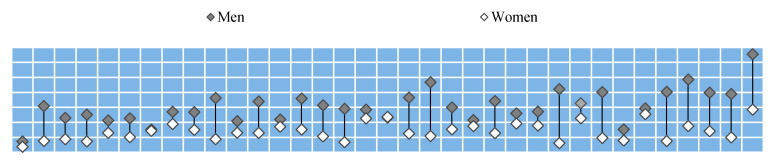
Standardized mortality rates due to lung cancer among Organization for Economic Cooperation and Development (OECD) countries, in 2017 or the latest available. Source: International Agency for Research on Cancer (IARC), GLOBOCAN 2018.

**Figure 2 ijerph-18-00447-f002:**
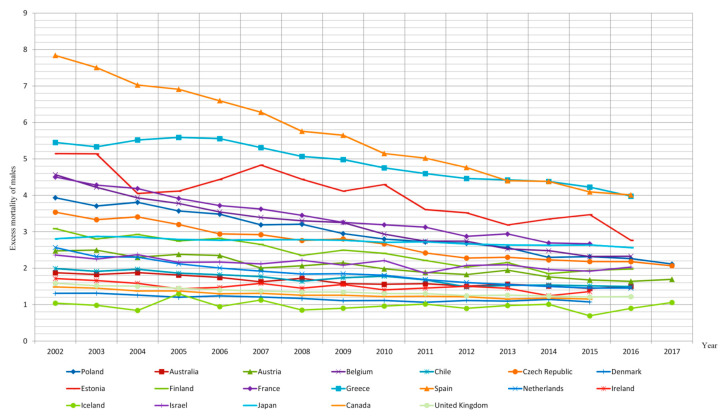
Excess mortality of males due to lung cancer in 18 selected OECD countries in 2002–2017.

**Figure 3 ijerph-18-00447-f003:**
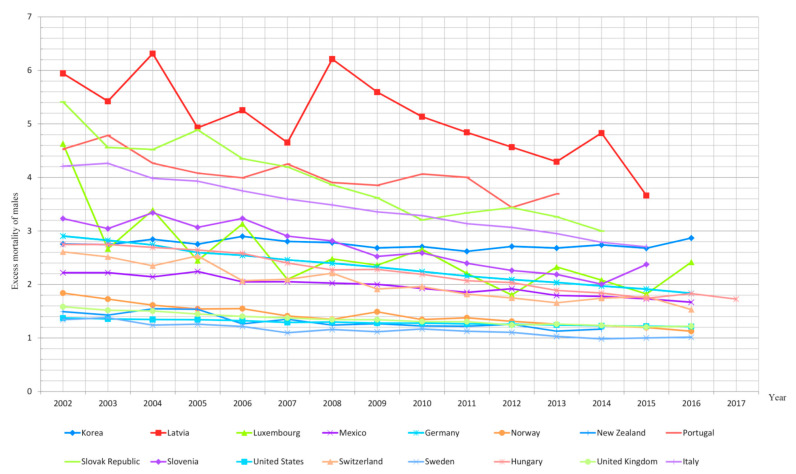
Excess mortality of males due to lung cancer in 17 selected OECD countries in the years 2002–2017.

**Figure 4 ijerph-18-00447-f004:**
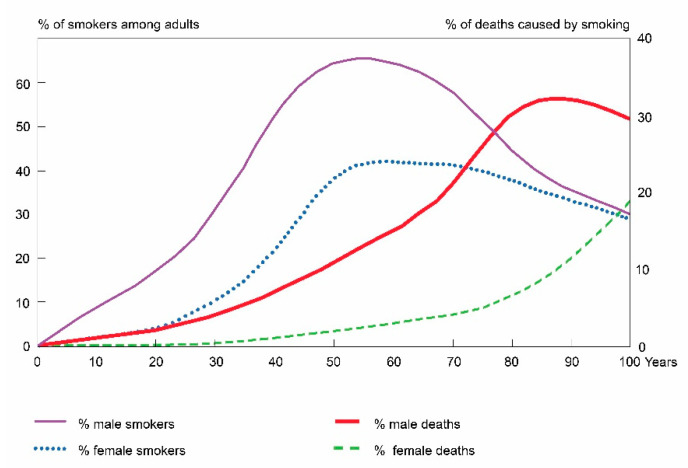
A model of an epidemic of nicotinism. According to Lopez et al.

**Table 1 ijerph-18-00447-t001:** Male excess mortality due to lung cancer in OECD countries from 2002 to 2017.

Country	Male Excess Mortality AAPC (95% CI)	Total Change in Male Excess Mortality (%)	AAPC of Male Mortality Rate (95% CI)	AAPC of Female Mortality Rate (95% CI)	Male Excess Mortality		Year
					2002	The Latest Year	
Spain *	−4.9 * (−5.1; −4.7)	−48.8	−0.2 * (−0.4; −0.1)	+4.9 * (+4.6; +5.2)	7.8	4	2016
Belgium *	−4.7 * (−5.0; −4.4)	−49.1	−1.7 * (−2.0; −1.3)	+3.2 * (+2.7; +3.8)	4.6	2.3	2016
Slovakia *	−4.4 * (−5.3; −3.5)	−44.6	−0.2 (−0.6; +0.2)	+4.4 * (+3.7; +5.1)	5.4	3	2014
Poland *	−4.1 * (−4.4; −3.9)	−46.3	−0.4 * (−0.6; −0.2)	+3.9 * (+3.6; +4.1)	3.9	2.1	2017
France *	−4.0 * (−4.3; −3.7)	−40.7	0.0 (−0.2; +0.1)	+4.1 * (+3.7; +4.5)	4.5	2.7	2015
Netherlands *	−3.9 * (−4.3; −3.5)	−43.4	−0.5 * (−0.8; −0.2)	+3.5 * (+3.1; +3.9)	2.6	1.5	2016
Luxembourg *	−3.8 * (−6.0; −1.6)	−47.8	−1.7 * (−2.9; −0.4)	+2.3 (−0.4; +5.0)	4.6	2.4	2016
Czech Republic *	−3.6 * (−4.0; −3.3)	−41.6	−1.7 * (−1.9; −1.5)	+2.0 * (+1.7; +2.4)	3.5	2.1	2017
Slovenia *	−3.6 * (−4.4; −2.7)	−26.5	−0.3 (−1.0; +0.4)	+3.4 * (+2.7; +4.0)	3.2	2.4	2015
Estonia *	−3.5 * (−4.6; −2.4)	−46.3	−0.6 (−1.3; +0.1)	+3.0 * (+2.1; +3.9)	5.1	2.8	2016
Switzerland *	−3.5 * (−4.1; −3.7)	−41.3	−1.1 * (−1.5; −0.7)	+2.5 * (+2.0; +3.0)	2.6	1.5	2016
Finland *	−3.5 * (−4.1; −2.9)	−36	−0.1 (−0.5; +0.3)	+3.5 * (+3.1; +4.0)	3.1	2	2016
Italy *	−3.5 * (−3.6; −3.3)	−35.8	−0.8 * (−1.0; −0.7)	+2.7 * (+2.5; +3.0)	4.2	2.7	2015
Hungary *	−3.4 * (−3.7; −3.1)	−33.3	0.0 (−0.3; +0.3)	+3.5 * (+3.1; 4.0)	2.7	1.8	2017
Germany *	−3.2 * (−3.3; −3.1)	−36.8	+0.2 * (0.0; +0.3)	+3.5 * (+3.3; +3.6)	2.9	1.8	2016
Norway *	−2.9 * (−3.4; −2.5)	−38.8	−1.2 * (−1.5; −0.9)	+1.8 * (+1.3; +2.3)	1.8	1.1	2016
Lithuania *	−2.6 * (−3.7; −1.5)	−38.4	+0.1 (−0.3; +0.5)	+2.8 * (+1.7; +3.8)	7.5	4.6	2017
Latvia *	−2.6 * (−4.0; −1.1)	−38.4	−0.5 (−1.1; +0.1)	+2.1 * (+0.7; +3.5)	6	3.7	2015
Greece *	−2.4 * (−2.8;−1.9)	−27.1	+1.8 * (+1.7; +2.0)	+4.3 * (+3.9; +4.7)	5.4	4	2016
New Zealand *	−2.4 * (−3.2; −1.5)	−21.9	−0.9 * (−1.5; −0.4)	+1.5 * (+0.8; +2.2)	1.5	1.2	2014
Chile *	−2.1 * (−2.6; −1.7)	−24.9	+1.4 * (+1.0; +1.9)	+3.6 * (+3.2; +4.0)	2	1.5	2016
Sweden *	−2.1 * (−2.6; −1.7)	−24.4	−0.7 * (−1.1; −0.4)	+1.4 * (+0.9; +1.9)	1.3	1	2016
Mexico *	−2.0 * (−2.3; −1.8)	−24.9	−2.0 * (−2.8; −1.2)	−0.2 (−0.7; +0.2)	2.2	1.7	2016
Portugal *	−2.0 * (−2.9; −1.1)	−18.4	+2.5 * (+2.0; +3.0)	+4.6 * (+3.9; +5.4)	4.5	3.7	2013
Austria *	−1.9 * (−2.2; −1.5)	−31.6	−0.3 * (−0.6; −0.1)	+2.6 * (+2.1; +3.0)	2.5	1.7	2017
Great Britain *	−1.9 * (−2.1; −1.6)	−23.1	−1.1 * (−1.2; −1.0)	+0.8 * (+0.5; +1.0)	1.6	1.2	2016
Canada *	−1.9 * (−2.2; −1.6)	−22.4	−0.7 * (−0.8; −0.5)	+1.2 * (+1.0; +1.4)	1.5	1.2	2015
Australia *	−1.9 * (−2.2; −1.5)	−20.1	−1.0 * (−1.3; −0.8)	+0.8 * (+0.5; +1.2)	1.9	1.5	2016
Ireland *	−1.5 * (−2.3; −0.8)	−20.9	−0.3 (−0.8; +0.2)	+1.3 * (+0.6; +1.9)	1.7	1.4	2015
Denmark *	−1.5 * (−2.0; −1.1)	−18	−0.6 * (−1.1; −0.1)	+0.9 * (+0.4; +1.5)	1.3	1.1	2015
Israel *	−1.2 * (−1.8; −0.7)	−14.2	+0.2 (−0.3; +0.6)	+1.4 * (+0.8; +2.0)	2.4	2	2016
US *	−0.9 * (−0.9; −0.8)	−11.6	−1.5 * (−1.6; −1.4)	−0.6 * (−0.8; −0.5)	1.4	1.2	2016
Japan *	−0.7 * (−0.8; −0.6)	−8.6	+1.9 * (+1.6; +2.2)	+2.6 * (+2.3; +2.9)	2.8	2.6	2016
Iceland	−0.8 (−2.4; +0.8)	1.9	−0.3 (−1.8; +1.2)	+0.5 (−0.6; +1.7)	1	0.9	2017
Korea	−0.2 (−0.5; +0.2)	4.1	+2.2 * (+2.0; +2.4)	+2.4 * (+2.1; +2.7)	2.8	2.9	2016

The relevant rows of Table 1 contain detailed information on total percentage change in excess mortality of males over the analyzed period, the year from which the most recent data on male and female mortality from lung malignant neoplasms in a given country are derived, male excess mortality from lung cancer in the first and last year of analysis and the average annual percentage change in male and female mortality from lung malignant cancers. * represents the significant values marked at *p* < 0.05.

## Data Availability

Data available in a publicly accessible repository that does not issue DOIs Publicly available datasets were analyzed in this study. This data can be found here: https://www-dep.iarc.fr/WHOdb/WHOdb.htm.

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
