# Peer review of "Excess Mortality of Males Due to Malignant Lung Cancer in OECD Countries"

_ijerph, 2021, doi:10.3390/ijerph18020447_

Round 1

Reviewer 1 Report

General comment

In the present study, authors focused on the changes in the excess mortality of males due to lung cancer and discussed the association with the changes in smoking rates among OECD countries. They concluded that the reduction in the excess mortality of males resulted from a decrease in male mortality and an increase in female mortality in many countries, and less frequently from a greater decrease in male mortality than female mortality due to pulmonary cancer. I would like to suggest minor revisions.

Minor comment

Introduction (line 34-37)

Authors stated as below.

  • life expectancy among women aged 0 increased from 48.1 to 70.5 years.
  • life expectancy among men increased from 52.9 to 75.6 years.

From the above, the life expectancy of men is 5.1 years longer than women in 2017. However, the authors next state that " In 2017, the difference in life expectancy between men and women worldwide was 5.1 years to the disadvantage of men. "

Please check if these notations are correct.

Discussion (line 217-219)

Authors explained that women's tobacco consumption decreased since the 1980s in the United States, Canada, Australia, the United Kingdom, Denmark, Sweden, but lung cancer mortality in women decreased slowly than men or rather increased.

Is there any information on the reason for this (eg, susceptibility associated with smoking-induced carcinogenesis)?

Author Response

Dear Sir/Madam

Thank you very much for your review and valuable comments. I have modified the content of the manuscript to implement your suggestions.

Please find my detailed replies to the individual passages of the article you have indicated. The changes were also made in the paper.

1.

Introduction (line 34-37)

Authors stated as below.

  • life expectancy among women aged 0 increased from 48.1 to 70.5 years.
  • life expectancy among men increased from 52.9 to 75.6 years.

From the above, the life expectancy of men was 5.1 years longer than women in 2017. However, the authors next state that " In 2017, the difference in life expectancy between men and women worldwide was 5.1 years to the disadvantage of men. "

Please check if these notations are correct.

The passage indicated by you has been changed in the text. Currently it takes the following form:

According to global data available for the years 1950-2017, life expectancy for men aged 0 increased from 48.1 to 70.5 years, whereas life expectancy for women increased from 52.9 to 75.6 years. In 2017, the difference in life expectancy between men and women worldwide was 5.1 years to the disadvantage of men.

  1.  

Discussion (line 217-219)

Authors explained that women's tobacco consumption decreased since the 1980s in the United States, Canada, Australia, the United Kingdom, Denmark, Sweden, but lung cancer mortality in women decreased slowly than men or rather increased.

Is there any information on the reason for this (eg, susceptibility associated with smoking-induced carcinogenesis)?

The exact mechanism explaining this phenomenon is unknown, but there are some hypotheses.

Boreham et al. in paper "Use of tobacco products. Health Survey for England. The Health of Minority Ethnic Groups ’99" have reported striking discrepancies, especially in women from some ethnic groups, between self-reported smoking habits and serum cotinine concentrations—a specific biomarker of nicotine absorption—suggesting that more women than men underreport their smoking habit.

Woodward M. et al. in paper "Smoking characteristics and inhalation biochemistry in the Scottish population" point out that women  might extract a greater quantity of carcinogens and other toxic agents from the same number of cigarettes than men.

Ryberg et al. have described higher DNA adduct levels among females than in males when adjusted for smoking dose. 

Kure, E.H.  et al. in their  paper"p53 mutations in lung tumours: relationship to gender and lung DNA adduct levels" indicates that relevant mutations which may be caused by tobacco carcinogens were found more frequently in women than in men. Studies have shown that TP53 is mutated in over two thirds of lung cancers, and that mutations are more frequently found in women than in men.

Reviewer 2 Report

The manuscript by Moryson and Stawinska-Witoszynska aims to assess time trends in the excess mortality of males due to lung cancer in OECD countries. The topic is interesting and fits the scope of the journal. The manuscript is written quite well; however, it has some fundamental flaws that prevent me from recommending it to publication in IJERPH.

1. First of all, the use of 'excess mortality' notion is wrong because what the Authors analyze is mortality 'sex-difference'. The term 'excess mortality' refers to '...the number of deaths actually recorded in excess of the number expected on the basis of past seasonal experience' (see Assaad et al. Bull World Health Organ. 1973; 49(3): 219–23 and several more recent studies, e.g. on COVID-19 excess mortality). Therefore, the figures you use do not describe the excess mortality of men. In your approach, the female mortality is treated as a reference while males' deaths represent 'excess' mortality. With this, you compare apples and oranges. The excess mortality of males could be represented by higher mortality rates of males compared to the mortality of males (not females) in other periods.

2. Secondly, the methodological approach used should be improved because what you do here is quite naive in methodological terms. The use of a linear trend for time-series data of 16 observations is not a robust method. Instead, you could use panel data regression with country-specific trends and data on tobacco use as an explanatory variable (and a set of socio-economic covariates). This would allow you to scrutinize the impact of tobacco on mortality and, at the same time, test the significance of trends itself. With your approach, it is questionable to point to an association between tobacco and mortality as you extensively do in your discussion. Your discussion is not much more than speculations. A more appropriate methodological approach would limit this speculativeness.

Other detailed concerns:

Abstract

3. lines 12-15: The sentence 'This study aimed to analyse changes in the level of the excess mortality of men caused by lung cancer ... in terms of ... mortality caused by this type of cancer' is partly a truism. See the parts that I gave in the above citation; is not it obvious that changes in excess mortality are analysed in terms of mortality?

4. lines 20-22: The sentence 'The decrease in excess mortality...' is illogic in light of my comment 1. Excess mortality of men cannot result from females mortality rates because this compares the phenomena that should not be compared in this context.

5. lines 22-24: Linguistic issue - the decline in male mortality cannot be observed in the female population.

Introduction

6. line 35: The figures you provide about the dynamics of LE in the period 1950-2017 are not correct and are in contrast with what you write just above. In lines 31-33, you state that women were expected to live longer at the beginning of the 20th century and these disparities in LE increased after WW2. Just after that, you state that during 1950-2017 period women LE increased from 48.1 to 70.5 while men LE from 52.9 to 75.6. This seems wrong because shows that both in 1950 and presently men have longer LE.

7. Items 8 and 9 are the same references; intext reference 10 does not refer to smoking; it seems that it is misplaced in text.

8. The aim of the study does not result from the content of your introduction; you do not give a reasonable argument of why lung cancer is of special interest. Also, you do not mention any previous research on the topic (while you do so in the discussion; lines 190-191) and do not show what is your original contribution to the state of science on this topic. Why your research is important, what is new about it is not clear.

Materials and methods

9. line 63: A specific reference for IARC data is missing.

10. This section requires extensive changes according to my comment 2.

Results

11.line 93-94: 'four Eastern and Southern European countries' is an imprecise statement that should be more explicit.

12. Data from figures 2 & 3 could be combined into a single, a page-sized large exhibit.

13. Again, if an alternative approach was used, this section should be rewritten extensively.

Discussion

14. Your comparison to other studies is very artificial, you only mention that these figures are consistent with previous reports which is not a decent comparison.

15. The sentences in lines 209-2131 'In Central...' and following is confusing. It does not reflect a consistent flow of logical argument.

16. In the sentence in lines 238-240 you explicitly state that you used another approach than used in many studies (crude vs. standardized coefficients) and give no explanation why you do so. This makes comparisons difficult and you provide no explanation of the superiority of your approach. In my opinion, the use of standardized data is more appropriate here because you compare populations of various countries and different points in time. This suffices for an argument for standardization. This is a serious shortcoming of your analysis.

17. Other study limitations are neglected while this part should definitely refer to methodological limitations at least.

To conclude, although the manuscript is written quite well, it has fundamental flaws that prevent me from recommending it to IJERPH readers.

Author Response

Dear Sir/Madam

Thank you very much for your review and valuable comments. I have modified the content of the article to implement you suggestions.

Please find my detailed replies to the individual passages of the article you have indicated. The changes were also made in the paper.

  1.  

First of all, the use of 'excess mortality' notion is wrong because what the Authors analyze is mortality 'sex-difference'. The term 'excess mortality' refers to '...the number of deaths actually recorded in excess of the number expected on the basis of past seasonal experience' (see Assaad et al. Bull World Health Organ. 1973; 49(3): 219–23 and several more recent studies, e.g. on COVID-19 excess mortality). Therefore, the figures you use do not describe the excess mortality of men. In your approach, the female mortality is treated as a reference while males' deaths represent 'excess' mortality. With this, you compare apples and oranges. The excess mortality of males could be represented by higher mortality rates of males compared to the mortality of males (not females) in other periods.

The use of the term excess mortality of males is commonly used by authors and defines a higher level of male mortality in relation to female mortality [1-6]. It is determined as a ratio of mortality rates in the male population to female mortality rates. It may be calculated for the total population as well as for individual age groups and it may describe mortality caused by various reasons. The phenomenon of excess male mortality is not new and concerns many countries. The finding of male excess mortality was confirmed with the introduction of official population statistics in all Western societies and has been documented since 1751 onward [1].

What we wanted, was to describe the phenomenon of excess male mortality due to lung cancer and its trends, and not to assess the changes in mortality of one sex, i.e. men.

  1. Luy M. Causes of Male Excess Mortality: Insights from Cloistered Populations. Population & Development Review. 2003;29(4):647–76.
  2. Maiolo V, Reid AM. Looking for an explanation for the excessive male mortality in England and Wales since the end of the 19th century. SSM - Population Health. 2020;11:100584.
  3. Beltrán-Sánchez H, Finch CE, Crimmins EM. Twentieth century surge of excess adult male mortality. Proc Natl Acad Sci USA. 2015;112(29):8993–8.
  4. French. F. E. Factors relating to the excess mortality of the male: A statistical investigation. PhD thesis. Univ. Calif. , Berkeley; 1960
  5. Herdan, G. Causes of excess male mortality in man. Acta Genet; 1952. 3:35 1-75
  6. Preston, S. H. An international comparison of excessive adult mortality. Populo Stud. 1970, 24:5-20

2.

Secondly, the methodological approach used should be improved because what you do here is quite naive in methodological terms. The use of a linear trend for time-series data of 16 observations is not a robust method. Instead, you could use panel data regression with country-specific trends and data on tobacco use as an explanatory variable (and a set of socio-economic covariates). This would allow you to scrutinize the impact of tobacco on mortality and, at the same time, test the significance of trends itself. With your approach, it is questionable to point to an association between tobacco and mortality as you extensively do in your discussion. Your discussion is not much more than speculations. A more appropriate methodological approach would limit this speculativeness.

Of course, we agree that other statistical methods could have been used, however, we chose a linear regression model because we did not analyse many variables. Our research was an epidemiological descriptive study in which it is impossible to assess the cause-and-effect relationship. The results of such studies allow to formulate hypotheses to be subsequently verified in analytical studies. Therefore, we did not set socioeconomic covariates and we could not investigate the effects of smoking on male mortality or excess mortality.

Our discussion is not a speculation but a assumption that the decrease in the number of smokers is the factor that contributes to the decline in mortality in general as well as the decline in excess mortality of men. This is acceptable because it has already been proven that smoking is the main contributor to the development of this type of cancer (about 80%).

3.

 lines 12-15: The sentence '

This study aimed to analyse changes in the level of the excess mortality of men caused by lung cancer ... in terms of ... mortality caused by this type of cancer' is partly a truism. See the parts that I gave in the above citation; is not it obvious that changes in excess mortality are analysed in terms of mortality?

The passage indicated by you has been changed in the text. Currently it takes the following form:

This study aimed to analyse the changes in the level of excess mortality of men caused by lung cancer between 2002-2017 in the countries associated with the Organization for Economic Cooperation and Development (OECD)

4.

lines 20-22: The sentence 'The decrease in excess mortality...' is illogic in light of my comment 1. Excess mortality of men cannot result from females mortality rates because this compares the phenomena that should not be compared in this context.

I would like to refer to our argument from point 1. In this context, we consider the phrase “The decrease in excess mortality ..” to be correct.

5.

 lines 22-24: Linguistic issue - the decline in male mortality cannot be observed in the female population.

The passage indicated by you has been changed in the text. Currently it takes the following form:

In most OECD countries the decrease in excess male mortality was the resultant of a decrease in mortality in the male population as well as the increase in female mortality. In 10 member countries of the OECD, there was a simultaneous increase in the mortality of men and women, though the increase was higher in the female population. Only in two countries the decrease in excess mortality of males was due to a greater decline in the mortality of men.

6.

line 35: The figures you provide about the dynamics of LE in the period 1950-2017 are not correct and are in contrast with what you write just above. In lines 31-33, you state that women were expected to live longer at the beginning of the 20th century and these disparities in LE increased after WW2. Just after that, you state that during 1950-2017 period women LE increased from 48.1 to 70.5 while men LE from 52.9 to 75.6. This seems wrong because shows that both in 1950 and presently men have longer LE.

The passage indicated by you has been changed in the text. Currently it takes the following form:

According to global data available for the years 1950-2017, life expectancy for men aged 0 increased from 48.1 to 70.5 years, whereas life expectancy for women increased from 52.9 to 75.6 years. In 2017, the difference in life expectancy between men and women worldwide was 5.1 years to the disadvantage of men.

7.

Items 8 and 9 are the same references; intext reference 10 does not refer to smoking; it seems that it is misplaced in text.

Item 9, like item 8, is an OECD report, but from another year, it has now been changed in the bibliography. Currently, the citation previously relating to item 10 is for item 9.

8.

The aim of the study does not result from the content of your introduction; you do not give a reasonable argument of why lung cancer is of special interest. Also, you do not mention any previous research on the topic (while you do so in the discussion; lines 190-191) and do not show what is your original contribution to the state of science on this topic. Why your research is important, what is new about it is not clear.

The introduction is supplemented with information why lung cancer is of special interest in terms of excess mortality of men. We also indicated previous research on the topic and pointed out that previous reports suggest that the analysis of excess mortality of men due to lung cancer is important and will make an original contribution to the state of knowledge on this topic.

The indicated passage (line 65-74) takes the following form:

The aim of the study was to analyse changes in the level of excess mortality of men due to lung cancer between 2002-2017 in the countries of the OECD. We considered this topic to be important because excess mortality of men concerns most diseases and deaths from external causes. However, despite the importance of the problem, the number of papers dealing with the subject seems to be inadequate. In view of increasing  significance of cancer related deaths, the topic of excessive mortality of males due to lung cancer, the most common cause of death among cancer patients, is of great importance. Especially so, since previous reports on changes in lung cancer mortality in the developed countries suggest changes in the socially and economically unfavourable phenomenon of excess mortality of males.

9.

 line 63: A specific reference for IARC data is missing.

IARC reference has been added

10.

Materials and methods

In the sentence in lines 238-240 you explicitly state that you used another approach than used in many studies (crude vs. standardized coefficients) and give no explanation why you do so. This makes comparisons difficult and you provide no explanation of the superiority of your approach. In my opinion, the use of standardized data is more appropriate here because you compare populations of various countries and different points in time. This suffices for an argument for standardization. This is a serious shortcoming of your analysis.

In order to compare male and female mortality in the OECD countries, the mortality rates standardized based on World Standardized population were used. Excess mortality of males was determined as the ratio of crude mortality rates in the male population to crude female mortality rates.

Figure 1 presently includes the standardized ratios.

11.

line 93-94: 'four Eastern and Southern European countries' is an imprecise statement that should be more explicit.

The passage indicated by you has been changed in the text. Currently it takes the following form:

The highest levels of male mortality were found in Eastern and Southern European countries, including Hungary, Greece, Poland, Slovakia and Slovenia followed by the Asian OECD member states.

  1.  

Data from figures 2 & 3 could be combined into a single, a page-sized large exhibit.

Figure 2 and 3 have been redesigned to provide better clarity. However, it was not possible to combine them into one graph while maintaining their legibility.

  1.  

Again, if an alternative approach was used, this section should be rewritten extensively.

The section on comparing mortality between different OECD countries has been revised to include standardized mortality rates.

14.

Your comparison to other studies is very artificial, you only mention that these figures are consistent with previous reports which is not a decent comparison

The passage indicated by you has been changed in the text. Currently it takes the following form:

The analysis of excess mortality trends presented in this publication has shown that the decrease in this phenomenon in most OECD countries results from a simultaneous increase in female mortality (33 out of 35 OECD countries) and a decrease in male mortality (25 out of 35 OECD countries). The results for the OECD countries are consistent with earlier reports of changes in mortality in several Western countries such as the United States, the United Kingdom, Canada and Australia. As early as 2010, Jemal A. et al. described the ongoing decline in male mortality and a plateau in female mortality since the end of the 20th century. [14]. In addition, in their analysis of 2013, Lortet-Tieulent J. et al. described a decrease in the incidence of lung cancer among men in 14 out of 26 European countries in the age group (35-64 years) and in 15 countries in the age group (65-74 years) between 1998 and 2007. In the female population, on the other hand, they observed the stabilisation of incidence of lung cancer among women in Eastern and Northern Europe, as well as an increase in lung cancer incidence in Western and Southern European countries. [15].

15.

The sentences in lines 209-2131 'In Central...' and following is confusing. It does not reflect a consistent flow of logical argument.

The section indicated by you has been changed in the text. Currently it takes the following form:

In Central European countries, the reduction in tobacco consumption only began in the 1980s, followed by a period when the rate of smoking reduction was at a similar level as in the countries where tobacco reduction had started earlier [24,32]. Therefore, over the period from 2002 to 2017, roughly 20 years since smoking had been declining in Central European countries and 50 years since the decline in smoking had been observed in North America, Australia and some of the countries of Western Europe, the reduction rates in male mortality in both groups remained on similar levels.

16.

Other study limitations are neglected while this part should definitely refer to methodological limitations at least.

The information about the methodological limitation has been included in the "Materials and Methods" section.

Our research was an epidemiological descriptive study in which it is impossible to assess the cause-and-effect relationship. The results of such studies allow for formulating hypotheses to be subsequently verified in analytical studies.

Reviewer 3 Report

Review of the manuscript Excess mortality of males due to malignant lung  cancer in OECD countries

Dear authors, thank you for this analysis. It is interseting to see that there is a decrease in mortality form lung cancer in males in several OECD countries.

I have some comments for you:

Line 16: AAPC means Annual Average Percent Change

Line 17-18: here you report that “A decrease in excess male mortality due to lung cancer between 2002 and 2017 was recorded in 33 of the 35 countries analysed”

And then  at lines 20-22 “The decrease in excess mortality of men in most OECD countries resulted from the simultaneous increase in female mortality (33 out of 35 OECD countries) and the decrease in male mortality (25 out of 35 OECD countries).

Could you explain better these data? They seem contradictory. First you say that males mortality decreased in 33 out of 35 countries, then in the second sentence is 25 out of 35. What are the mortality causes here for females and male, lung cancer only?

Lines 28-29: It is not clear the link between the first sentence and the second. One says that on average the male life expectancy at birth is shorter than that of females. The second one says that this is due to the fact that the rate of male deaths is higher than females. But this second sentence does not appear to be related to the first one. They are two separate facts. What does it mean that because the rate of male deaths are higher than females this explain why male life expectancy at birth is shorter than females? Please explain in a clear way.

Moreover, it would be better to report a more recent bibliographic reference about these data (you mention a study of 14 years ago, is this confirmed by 2020?).

Line 35: these data are not correct.  You say that women life expectancy increased from 48.1 to 70.5; while males from 52.9 to 75.6. So according to what you write the men have a higher life expectancy. But you say that women have 5.1 higher expectancy compared to men.

Lines 80-82: I am not an expert of regression models, this should be reviewed by an expert  statistician, I cannot give any judgment.

Figure 2 and 3: could please make them more clear (for example the colours or widen the figure)? Since there are so many countries the trends are a bit difficult to distinguish

Line 50: has only lung cancer been the reason of mortality increase?

In general: I would like more considerations explaining why the women mortality increased and why decreased the male mortality, which conditions and lifestyles changed and what changed in several countries (for example in Belgium, Spain and Luxembourg and Central Eastern Europe and also for other non European countries)? Is it all due exclusively to smoking reduction/increase? And what about working conditions, environmental pollution or type of places where people are living?

 Why has Portugal still high excess mortality in males? Why in Korea mortality rates of females for lung cancer is so high?

Author Response

Dear Sir/Madam

Thank you very much for your review.

I consider your remarks very valuable. I haveanalysed them with great diligence and modified the content of the manuscript to implement them. I believe that thanks to the advice provided, the work becomes clearer and more valuable for the readers.

Please find my detailed replies to the individual passages of the article you have indicated. The changes were also introduced in the paper.

1.

Line 16: AAPC means Annual Average Percent Change

The abbreviation in line 16 (AAPC) has been fully developed and now the sentence takes the following form:

In order to compare changes in male mortality rates across countries, the Average Annual Percent Change (AAPC) in male excess mortality rate for a given country was calculated.

2.

Line 17-18: here you report that “A decrease in excess male mortality due to lung cancer between 2002 and 2017 was recorded in 33 of the 35 countries analysed”

And then  at lines 20-22 “The decrease in excess mortality of men in most OECD countries resulted from the simultaneous increase in female mortality (33 out of 35 OECD countries) and the decrease in male mortality (25 out of 35 OECD countries).

Could you explain better these data? They seem contradictory. First you say that males mortality decreased in 33 out of 35 countries, then in the second sentence is 25 out of 35. What are the mortality causes here for females and male, lung cancer only?

The passage indicated by you has been changed in the text. Currently it takes the following form:

A decrease in excess male mortality due to lung cancer between 2002 and 2017 was recorded in 33 of the 35 countries analysed. In most OECD countries the decrease in excess male mortality was the resultant of a decrease in mortality in the male population as well as the increase in female mortality. In 10 member countries of the OECD, there was a simultaneous increase in the mortality of men and women, though the increase was higher in the female population. Only in two countries the decrease in excess mortality of males was due to a greater decline in the mortality of men.

3.

Lines 28-29: It is not clear the link between the first sentence and the second. One says that on average the male life expectancy at birth is shorter than that of females. The second one says that this is due to the fact that the rate of male deaths is higher than females. But this second sentence does not appear to be related to the first one. They are two separate facts. What does it mean that because the rate of male deaths are higher than females this explain why male life expectancy at birth is shorter than females? Please explain in a clear way.

Moreover, it would be better to report a more recent bibliographic reference about these data (you mention a study of 14 years ago, is this confirmed by 2020?).

The source of information about the longer life expectancy of women has been replaced by the 2019 WHO report (World health statistics overview 2019: monitoring health for the SDGs, sustainable development goals. Geneva: World Health Organization; 2019).

The link between the first and second sentence has been also changed. Currently it takes the following form:

Presently, the average male life expectancy at birth is shorter than that of females. This phenomenon is caused by the prevalence of male deaths over female deaths.

4.

Line 35: these data are not correct.  You say that women life expectancy increased from 48.1 to 70.5; while males from 52.9 to 75.6. So according to what you write the men have a higher life expectancy. But you say that women have 5.1 higher expectancy compared to men.

The fragment indicated by you has been changed in the text. Currently it takes the following form:

According to global data available for the years 1950-2017, life expectancy for men aged 0 increased from 48.1 to 70.5 years, compared to that for women which increased from 52.9 to 75.6 years. In 2017, the difference in life expectancy between men and women worldwide was 5.1 years to the disadvantage of men.

5.

Figure 2 and 3: could please make them more clear (for example the colours or widen the figure)? Since there are so many countries the trends are a bit difficult to distinguish

Figure 2 and 3 have been redesigned to provide better clarity.

  1.  

Line 50: has only lung cancer been the reason of mortality increase?

The changes in lung cancer mortality are not the sole reason for the decrease of the gender gap in life expectancy. Louise Sundberg et al. in their paper "Why is the gender gap in life expectancy decreasing? The impact of age- and cause-specific mortality in Sweden 1997–2014" summed up the effects of individual diseases on gender gap in life expectancy. According to this study, decreased mortality from ischemic heart disease had the largest impact on the increased life expectancy of both men and women and on the decreased gender gap in life expectancy. The increased mortality from Alzheimer’s disease negatively influenced overall life expectancy, but because of higher female mortality, it also served to decrease the gender gap in life expectancy. The impact of other causes of death, particularly smoking-related causes, decreased in men but increased in women, also reducing the gap in life expectancy.

7.

In general: I would like more considerations explaining why the women mortality increased and why decreased the male mortality, which conditions and lifestyles changed and what changed in several countries (for example in Belgium, Spain and Luxembourg and Central Eastern Europe and also for other non European countries)? Is it all due exclusively to smoking reduction/increase? And what about working conditions, environmental pollution or type of places where people are living?

Our research was an epidemiological descriptive study in which it is impossible to assess the cause-and-effect relationship.The result of such studies allow to formulate hypotheses to be subsequently verified in analytical studies. Therefore, we cannot determine the exact relationship between the decline in mortality and such factors as lifestyle changes, working conditions, environmental pollution or type of places where people are living. These factors are very likely to be related to thereduction of the excess male mortality, which we considered in the discussion of our paper. However, our discussion is only an assumption that a change in the prevalence of smoking  may have contributed to the decline in male and female mortality. This is acceptable as it has already been proven that smoking is the main contributor to the development of this type of cancer (about 80%).

8.

Why has Portugal still high excess mortality in males?

Carreira et al. in the paper “Trends in the prevalence of smoking in Portugal: a systematic review.” BMC Public Health, 2012,point out that:

The gender difference in the mortality due to lung cancer in Portugal  is likely to be explained by the fact that Portuguese women are at earlier stages of the smoking epidemic than men. According to the epidemic model, among women the current stage is characterized by a rapid increase of the prevalence of smoking, along with few deaths attributable to smoking.

9.

Why in Korea mortality rates of females for lung cancer is so high?

Park et al. in the paper of Lung Cancer in Korea: Recent Trends. Tuberc Respir Dis2016, indicate that:

The burden of lung cancer among women in Korea is high due to increasing incidence of lung cancer in never-smokers. The World Health Organization estimates that among the lung cancer cases worldwide, 25% were never-smokers. However, according to In et al., the proportion of lung cancer occurring in never-smokers in Korea  is different.  In the group of 8,788 Korean patients (female, 24.2%) diagnosed in 2005, 87.3% of males were current or former smokers whereas 79.7% of women had no history of smoking.

The etiology of lung cancer occurring in never-smokers remains uncertain although several possible risk factors have been suggested, including environmental tobacco smoke, occupational carcinogens exposure, cooking fume exposure, oncogenic virus, pre-existing lung diseases, diet, and estrogen.

Round 2

Reviewer 2 Report

  1. Apparently, the term 'excess mortality' is not unambiguous and has two meanings; the one you used and the one I mentioned. As such, it requires clear delimitation and definition along with mentioning that the term has other meanings. Unfortunately, what you did was staying on your position, ignoring my comment in the manuscript and instructing me that mortality sex-difference is a well-known phenomenon. Well, I guess I would not be asked to review your manuscript if I was not aware of this demographical truism.
  2. Your response to my concern regarding methodological issues is surprising. You contradict yourself by stating that your purpose was not to establish a causal relationship while in the whole discussion you investigate this relationship. Regardless of this, in my opinion, your approach is limited and - even if your descriptive analysis is accepted - use of time-series analysis for 16 yearly observations is not justified (you have not responded to this issue in your response).
  3. You state that the purpose of your study was descriptive '...in which it is impossible to assess the cause-and-effect relationship'. That is the point. If you have other available ways to investigate the scientific problem (I gave an example in the 1st round) which are more robust and could provide more answers to a scientific problem, you are expected to use such a more exhaustive approach. Therefore, I do not consider your way of doing this research as a robust and satisfying way of solving a research question.
  4. You state that 'The results of such studies allow for formulating hypotheses to be subsequently verified in analytical studies'. I would be happy to see what this hypothesis would be because you do not formulate any.

Author Response

Dear Sir/Madam

            Thank you for your comment.

            I would like to apologise to you for  the misunderstanding that has arisen. I am truly grateful for your comments and I have analysed all of them thoroughly. I am very sorry if my response to your previous remarks was perceived as an attempt to instruct you. That was not my intention. I just wanted to explain my view of the work you were analysing. 

            Below please find my detailed response to your remarks.

  1. Apparently, the term 'excess mortality' is not unambiguous and has two meanings; the one you used and the one I mentioned. As such, it requires clear delimitation and definition along with mentioning that the term has other meanings. Unfortunately, what you did was staying on your position, ignoring my comment in the manuscript and instructing me that mortality sex-difference is a well-known phenomenon. Well, I guess I would not be asked to review your manuscript if I was not aware of this demographical truism.

            The information on the double meaning of the phrase 'male excess mortality' has now been included in the text of the work.

Line 84-89:  

Excess mortality of males was determined as the ratio of crude mortality rates in the male population to crude female mortality rates. It should be noted that the phrase 'male excess mortality' may also have a different meaning. While in this work it means the excess deaths of men over deaths of women due to the same cause, the phrase may also mean the number of male deaths recorded in excess of the number of male deaths expected on the basis of past seasonal experience.

  1. Your response to my concern regarding methodological issues is surprising. You contradict yourself by stating that your purpose was not to establish a causal relationship while in the whole discussion you investigate this relationship. Regardless of this, in my opinion, your approach is limited and - even if your descriptive analysis is accepted - use of time-series analysis for 16 yearly observations is not justified (you have not responded to this issue in your response).

In this work, we have carried out an epidemiological analysis which does not aim to demonstrate the link between smoking and mortality caused by malignant lung cancers. In the discussion section of our paper, I refer to the publications in which the correlation between smoking and lung cancer mortality was examined very thoroughly. In addition, I describe and refer to a previously developed model of the smoking epidemic and the time shift between the prevalence of smoking and mortality due to malignant lung cancers. Indeed, the method of panel data regression you referred to is an interesting statistical tool of great relevance to epidemiological studies, yet our aim was not to investigate the link between lung cancer mortality trends and tobacco consumption figures or other socio-economic data. Our work is focused on describing the changes that we believe to be an important phenomenon, namely, the excess mortality rate of men in the broad time span of changes in the spread of nicotinism.

            An important part of our work involves comparing the changes in mortality in the OECD countries which, though they are currently adopting a very similar approach in terms of smoke-free policies, differed enormously until the late 1980s. Although our work analyses the changes in mortality after 2002, when male excess mortality from lung cancer could be seen both in the countries that only acceded to the OECD in the 1990s and in countries that had previously joined the OECD, we consider it crucial to be able to put the changes occurring in the 21st century in the context of the differences in the course of the epidemic of smoking since the 1960s.

  1. You state that the purpose of your study was descriptive '...in which it is impossible to assess the cause-and-effect relationship'. That is the point. If you have other available ways to investigate the scientific problem (I gave an example in the 1st round) which are more robust and could provide more answers to a scientific problem, you are expected to use such a more exhaustive approach. Therefore, I do not consider your way of doing this research as a robust and satisfying way of solving a research question.

            Our work was not intended to identify the link between smoking and the problem of mortality from malignant lung cancer. This subject had already been thoroughly researched and described, and the conclusions of the authors of these studies are referred to in our publication. In our paper, we describe changes in mortality from lung cancer and place them in the broad perspective of changes in the spread of nicotinism whose time frame goes beyond the period in which we examine the changes in mortality rates. We find this approach interesting and - in the discussion section - we justify its validity using the model of the epidemic of nicotinism according to Lopez et al.

  1. You state that 'The results of such studies allow for formulating hypotheses to be subsequently verified in analytical studies'. I would be happy to see what this hypothesis would be because you do not formulate any.

The intention of the wording I used was to point out that in the paper I describe the trends in the excess mortality of men and then juxtapose them with the trends in the prevalence of smoking. The juxtaposition is based on the cause and effect relationship between smoking and mortality from malignant lung cancer described in other authors' works.

Thank you for drawing my attention to this paragraph. It has now been amended and takes the following form:

Line 102-108:

This work is designed as an epidemiological descriptive study. Its aim is to describe the trends in the excess mortality of men in OECD countries and to compare them with the trends in the prevalence of smoking in individual countries. The methodology used does not allow to show a cause-and-effect correlation between these factors. For this reason, the juxtaposition of lung cancer mortality with the prevalence of smoking is based on the results of previous analytical studies, which are cited in the subsequent section of this paper.

Round 3

Reviewer 2 Report

Please make sure if the asterisks at the countries' names are needed.